# Trends in Botulinum Toxin Use among Patients with Multiple Sclerosis: A Population-Based Study

**DOI:** 10.3390/toxins15040280

**Published:** 2023-04-12

**Authors:** Djamel Bensmail, Pierre Karam, Anne Forestier, Jean-Yves Loze, Jonathan Lévy

**Affiliations:** 1Department of Physical and Rehabilitation Medicine, Raymond-Poincaré Teaching Hospital, APHP, Université Paris-Saclay, 92380 Garches, France; jonathan.levy2@aphp.fr; 2Unité INSERM 1179, University of Versailles Saint-Quentin-en-Yvelines, 78180 Montigny-Le-Bretonneux, France; 3PKCS, 69130 Ecully, France; p.karam@pkcs.fr; 4Ipsen, 92100 Boulogne-Billancourt, France; anne.forestier@ipsen.com (A.F.); jean-yves.loze@ipsen.com (J.-Y.L.)

**Keywords:** botulinum toxin, France, multiple sclerosis, spasticity, neurogenic detrusor overactivity

## Abstract

There are limited real-world data on the use of botulinum toxin type A (BoNT-A) in patients with multiple sclerosis (MS). Accordingly, this nationwide, population-based, retrospective cohort study aimed to describe BoNT-A treatment trends in patients with MS between 2014 and 2020 in France. This study extracted data from the French National Hospital Discharge Database (*Programme de Médicalisation des Systèmes d’Information*, PMSI) covering the entire French population. Among 105,206 patients coded with MS, we identified those who received ≥1 BoNT-A injection, administered within striated muscle for MS-related spasticity and/or within the detrusor smooth muscle for neurogenic detrusor overactivity (NDO). A total of 8427 patients (8.0%) received BoNT-A injections for spasticity, 52.9% of whom received ≥3 BoNT-A injections with 61.9% of the repeated injections administered every 3 to 6 months. A total of 2912 patients (2.8%) received BoNT-A injections for NDO, with a mean of 4.7 injections per patient. Most repeated BoNT-A injections within the detrusor smooth muscle (60.0%) were administered every 5 to 8 months. There were 585 patients (0.6%) who received both BoNT-A injections within striated muscle and the detrusor smooth muscle. Overall, our study highlights a broad range of BoNT-A treatment practices between 2014 and 2020 in patients with MS.

## Plain Language Summary

Botulinum toxin type A (BoNT-A) is an injectable muscle relaxant that is used to treat spasticity, defined as abnormal muscle tightness due to prolonged muscle contraction. BoNT-A is also used to treat neurogenic detrusor overactivity (NDO), defined as involuntary bladder contractions. Both spasticity and NDO are common symptoms of multiple sclerosis (MS), a chronic disease that damages the nerve fibres in the brain and spinal cord.

In this nationwide study from France, we described the real-world use of BoNT-A during 2014–2020 based on the electronic health records of patients with MS suffering from spasticity and NDO.

Among a total of 105,206 patients diagnosed with MS between 2014 and 2020, 8% of the patients received BoNT-A injections into the affected muscles to treat spasticity, 3% received BoNT-A injections into the bladder to treat NDO, and <1% received both types of BoNT-A injections. More than half of the BoNT-A-treated patients received repeated injections of BoNT-A, at an interval of 3–6 months for injections into muscles and 5–8 months for injections into the bladder. This is in line with the known duration of action of BoNT-A, which offers symptom relief beyond 3 months for spasticity and beyond 8 months for NDO.

In summary, this study highlights a broad range of BoNT-A treatment practices in patients with MS and reinforces the importance of repeated BoNT-A injections to provide good control of spasticity and NDO.

## 1. Introduction

Botulinum toxin type A (BoNT-A) is a powerful neurotoxin that acts by inhibiting the presynaptic transmission of acetylcholine at the neuromuscular junction [1]. Given its potency and favourable safety profile, BoNT-A is routinely used for the treatment of neurologic movement disorders, including dystonia [2,3]. BoNT-A is also used for the treatment of spasticity and neurogenic detrusor overactivity (NDO) [4]. Both spasticity and NDO are common secondary conditions in multiple sclerosis (MS), affecting up to 80% and 60% of these patients, respectively [1,5].

The treatment of MS-related spasticity usually combines physiotherapy with pharmacological interventions, including baclofen (oral or intrathecal administration), tizanidine, dantrolene, benzodiazepines (e.g., clonazepam and diazepam), cannabinoids, and intramuscular BoNT-A injection [1,6,7]. However, evidence-based clinical practice guidelines consistently recommend the use of BoNT-A as a first-line treatment of MS-related focal spasticity [8,9]. Patients with spasticity usually require repeated BoNT-A injections within striated muscles once every 3 to 4 months [10].

Injections of BoNT-A within the detrusor smooth muscle are also currently recognised as the standard treatment for NDO among patients with MS when antimuscarinics reveal to be ineffective or poorly tolerated [11,12]. By the blockade of the neurotransmitter (acetylcholine) release and suburothelial sensory receptors’ expression, BoNT-A could cause chemical denervation of the detrusor muscle, with clinical benefits lasting for up to 9 months [12].

Despite the potential therapeutic benefits of BoNT-A in patients with MS, there are limited real-world data on its use in this patient population. Moreover, in countries such as the United States and Canada, BoNT-A use may be limited by patient costs and insurance coverage [13]. This is, however, not the case in France, where patients with MS are covered for 100% of their healthcare costs [14]. In this context, the present study, based on the real-world analysis of the French National Hospital Discharge Database (*Programme de Médicalisation des Systèmes d’Information*, PMSI) covering the entire French population, aims to describe the trends in BoNT-A use among patients with MS between 2014 and 2020.

## 2. Results

### 2.1. Study Population

A total of 105,206 patients coded with MS were identified between 2014 and 2020, including 74,382 women (70.7%) and 30,824 men (29.3%), resulting in a female-to-male sex ratio of 2.4. Most patients with MS belonged to the age group 40–59 years (*n* = 46,030; 43.8%) followed by 20–39 years (*n* = 29,397; 27.9%) (Figure 1). Among the patients with MS, spasticity was identified in 24,878 patients (23.6%). In general, men were more likely to be coded with MS-related spasticity than their female counterparts (29.9% versus 21.0%; *p* < 0.0001). The prevalence of MS-related spasticity was found to progressively increase with age, peaking in the 60–79-year age group at 31.3%, before decreasing thereafter (Figure 2A).

### 2.2. BoNT-A for MS-Related Spasticity

A total of 8427 patients with MS (8.0%) received at least one BoNT-A injection between 2014 and 2020 within striated muscles for the treatment of spasticity. More than half of these BoNT-A-treated patients (4461/8427; 52.9%) received at least three BoNT-A injections. Men with MS were more likely to be treated with BoNT-A for spasticity (10.8% versus 6.8%; *p* < 0.0001). Regarding age, the highest rate of BoNT-A use for spasticity among patients with MS was reported in the 40–59-year age group (10.3%) (Figure 2A).

In total, there were 42,147 BoNT-A injections for spasticity recorded in the PMSI database among the patients with MS. There was a steady increase in the number of administered BoNT-A injections within striated muscles between 2014 and 2019 (Figure 3A). The mean ± standard deviation (SD) interval between two BoNT-A injections within striated muscles was 6.0 ± 6.0 months. As illustrated in Figure 4A, most repeated BoNT-A injections (61.9%) for spasticity were administered every 3 to 6 months.

When excluding those who initiated BoNT-A therapy before 2015 to evaluate the trends of BoNT-A initiation more accurately for MS-related spasticity, it was revealed that 54.6% of the patients with MS-related spasticity received at least three BoNT-A injections between 2015 and 2020 (Figure 5A). Moreover, the rate of BoNT-A use was estimated at 38.0% among the patients with MS who initiated BoNT-A therapy for spasticity in 2015, received at least three injections, and who were still being treated with BoNT-A in December 2020. Among those who discontinued the BoNT-A therapy before December 2020, the mean ± SD duration of the BoNT-A therapy was 33.6 ± 17.5 months. The estimated rate of the BoNT-A treatment initiation among the patients hospitalised for the first time for MS between 2017 and 2020 was stable, ranging from 17.1% to 18.0% (Table 1). 

### 2.3. BoNT-A for Neurogenic Detrusor Overactivity

Among a total of 105,206 patients coded with MS, 2912 (2.8%) received BoNT-A injections for NDO between 2014 and 2020. There was a total of 13,725 BoNT-A injections administered into the detrusor muscle between 2014 and 2020. The administration of BoNT-A for NDO was mostly performed in large medical centres located in the Paris region and Rhône-Alpes (southeast region of France) (Appendix A). The rate of BoNT-A use for NDO followed a similar age pattern as the rate of BoNT-A use for spasticity, as the highest rate of BoNT-A use for NDO among patients with MS was reported in the 40–59-year age group (3.6%) (Figure 2B).

A steady increase in the number of BoNT-A injections within the detrusor muscle was reported between 2014 and 2019 (Figure 3B). Most repeated BoNT-A injections for NDO (60.0%) were administered every 5 to 8 months (Figure 4B), with a mean ± SD interval between two BoNT-A injections of 8.3 ± 5.0 months. The mean ± SD number of BoNT-A injections per patient was 4.7 ± 3.7, with 59.3% of the 2912 patients receiving at least three BoNT-A injections within the detrusor muscle (Figure 5B).

Overall, there were 585 patients (0.6%) who received both BoNT-A injections within striated muscles and the detrusor smooth muscle, of whom 299 patients received both types of injections in an overlapping or alternating manner within one-year intervals.

## 3. Discussion

There are currently very limited published data on the clinical use of BoNT-A therapy in patients with MS [15]. Our 7-year longitudinal analysis of the comprehensive French PMSI database extends the literature on this aspect by highlighting a broad range of BoNT-A treatment practices between 2014 and 2020 in this patient population.

The present study included a total of 105,206 patients with MS, 23.6% of whom were coded as having spasticity. This figure may be lower compared to other prevalence studies [16,17,18]. In a cross-sectional, registry-based study from the United States conducted in 20,380 patients with MS, 84% of the patients experienced some form of spasticity, with the severity ranging from minimal (31%) to total (4%) [16]. Similar results were obtained in a cross-sectional survey study from the United Kingdom, which reported spasticity in 85.4% of a total of 701 patients with MS [18]. In another cross-sectional, survey-based study from Spain performed in 2029 patients with MS, the prevalence of MS-related spasticity was 65.7%, with the severity ranging from mild (59.9%) to severe (22.7%) [17]. The lower spasticity prevalence reported in our longitudinal analysis compared to other studies may be the result of different study settings and study methodologies, as well as study populations with spasticity presenting with varying degrees of severity. Most importantly, unlike other prevalence studies [16,17,18] in which spasticity was assessed using self-report questionnaires, spasticity in the present study was identified by International Classification of Diseases version 10 (ICD-10) coding made by the treating physicians. Hence, it is likely that the spasticity identified through medical coding in the present analysis may be disabling, requiring pharmacological treatment.

In the current study, the rate of BoNT-A use for spasticity in patients with MS was 8.0%, with men being more likely to be treated with BoNT-A than women (10.8% versus 6.8%). This is consistent with an observation that female patients with MS-related spasticity are less likely to experience severe spasticity compared to their male counterparts (odds ratio, 0.83; 95% confidence interval, 0.75–0.93) [16]. The rate of BoNT-A use for NDO in patients with MS was even lower in the present study at 2.8%. Logistical constraints may be behind the reduced use of BoNT-A in this study, including a lack of access to specialists performing intradetrusor BoNT-A injections. Indeed, these logistical constraints were reflected in the present analysis by the concentration of BoNT-A injections into the detrusor muscle in large medical centres located in urban areas, whereas BoNT-A administration was limited in smaller healthcare facilities and in rural/semi-rural areas.

Owing to the reversible pharmacological effect of BoNT-A, repeated BoNT-A injections are necessary to maintain a satisfactory therapeutic outcome against MS-related spasticity and NDO [19]. Accordingly, in this study, 52.9% of the BoNT-A-treated patients received at least three intramuscular BoNT-A injections for MS-related spasticity, with 61.9% of these repeated BoNT-A injections administered every 3 to 6 months. This observation is in line with the general recommended interval of 3–4 months between two BoNT-A injections administered within striated muscles [10] as well as recent pharmacological studies [20].

Similarly, in the current analysis, the mean number of BoNT-A injections for NDO per patient was 4.7. Overall, 60.0% of the repeated BoNT-A injections into the detrusor muscle were administered every 5 to 8 months, which is consistent with the mean duration of efficacy of BoNT-A for NDO estimated at 7−8 months [21]. In line with our study, Joussain and colleagues [22] performed a retrospective analysis of 292 patients with spinal cord injury or MS who had received BoNT-A injections for NDO. The mean ± SD number of BoNT-A injections was 9.7 ± 4.7 per patient, and the mean interval between two injections ranged from 5.1 to 6.5 months [22]. Other notable findings of the aforementioned study [22] were low rates of BoNT-A treatment withdrawal after 7 years of follow-up (mainly due to difficulties related to clean intermittent catheterisation and personal convenience) (11.3%) and of primary failure, defined as the absence of efficacy after the first BoNT-A injection (5.1%), with 60.8% of the followed-up patients at 7 years still treated with BoNT-A injections [22].

Although the long-term efficacy and safety of BoNT-A injections for NDO have been consistently demonstrated [19,21,22,23], there is evidence of decreasing efficacy over time for intradetrusor BoNT-A therapy, particularly after the fourth injection [19,23]. This further reinforces the importance of repeated BoNT-A injections to treat detrusor overactivity and improve bladder compliance in patients with NDO.

It has been also suggested that the clinical use of BoNT-A in patients with MS largely depends on symptom burden rather than merely being a function of the disease duration [13]. Moreover, spasticity may worsen with muscle fatigue, stress, and anxiety [24]. MS-related spasticity also worsens with greater gait disability, as well as with relapses or the progression of the disease, treatments for the disease process, and treatments for other symptoms, infections, injuries, or wounds [25]. As such, BoNT-A injections could be scheduled depending on symptom burden as well as the intrinsic and extrinsic factors that can influence spasticity in people with MS. Hence, improving the understanding of spasticity triggers and of MS-specific clinical features may enable clinicians to better individualise BoNT-A therapy [24]. The routine assessment of urodynamic parameters (e.g., maximum detrusor pressure, bladder capacity, bladder compliance, and non-voiding contractions) is also important for the individualisation of intradetrusor BoNT-A therapy and for improving therapeutic compliance [22,23]. Overall, individualised and flexible treatment intervals of BoNT-A tailored to individual clinical needs may be important to reduce patient and carer burden [26,27,28].

There were limitations to the present study, inherent to retrospective analyses of electronic health records, including potential bias, missing data, and coding errors, as coding was made by the treating physicians and hence relies solely on physicians’ accuracy in selecting medical codes. There is also a potential for the misclassification of MS disease status. Furthermore, we did not have access to the data regarding BoNT-A dosage or effectiveness. In addition, this study did not examine the correlation between BoNT-A use and clinical and demographic variables, because the use of the anonymised PMSI database does not allow the performance of a cross-check analysis of clinical data. Given the impact of spasticity and NDO on the day-to-day lives of people with MS, studying the clinical and demographic factors that may influence BoNT-A treatment patterns is warranted. Lastly, we did not evaluate BoNT-A use according to MS subtypes (i.e., relapsing remitting MS, secondary progressive MS, and primary progressive MS), as the French version of ICD-10 does not make the distinction between the different MS subtypes.

Our study is however strengthened by the use of the PMSI national database covering the entire population of France (over 66 million people), which enabled the inclusion of almost all the patients with MS in France, thus minimising the risk for selection bias. Another strength of this study is the long observation period with exhaustive data collection on the trends in the use of BoNT-A, administered both within striated muscle and the detrusor smooth muscle. To the best of our knowledge, this is the first reported large analysis exploring intradetrusor BoNT-A use in patients with MS. Moreover, this study is one of the few studies to have used a comprehensive national database to describe BoNT-A use for MS-related spasticity.

## 4. Conclusions

In a contemporary population from France of 105,206 patients coded with MS between 2014 and 2020, 8427 (8.0%) received BoNT-A injections for the treatment of spasticity and 2912 (2.8%) received BoNT-A injections for NDO. In adherence with BoNT-A treatment recommendations, more than half of the BoNT-A-treated patients received repeated injections of BoNT-A at an interval of 3–6 months for injections within striated muscles and 5–8 months for intradetrusor injections. Our study reinforces the importance of repeated BoNT-A injections to maintain a satisfactory therapeutic outcome against MS-related spasticity and NDO.

## 5. Materials and Methods

### 5.1. Study Design and Data Source

This was a nationwide, population-based, retrospective cohort study, using data extracted between 2014 and 2020 from the PMSI database, which covers the whole French population, corresponding to over 66 million persons. The PMSI database contains individualised and comprehensive data on all hospital stays in conventional medical units, rehabilitation units, homecare units, and psychiatric institutions in France. The PMSI database also contains patient information on medical diagnoses, costly drugs, and implantable medical devices administered during each hospital stay. In the present study, medical diagnoses were coded according to the ICD-10 and medical procedures according to the French Joint Classification of Medical Procedures (*Classification Commune des Actes Médicaux*, CCAM).

### 5.2. Study Population and Botulinum Toxin Injections

Patients diagnosed with MS, throughout France, between 2014 and 2020 were included in this study. The ICD-10 code used for MS diagnosis was “G35”. Among patients coded with MS, we searched the PMSI database to identify those with MS-related spasticity, using the following ICD-10 codes: “G81.1”, “G82.1”, and “G82.4”. We also searched the PMSI database to identify patients coded with MS who received at least one BoNT-A injection for spasticity between 2014 and 2020, using the following CCAM codes: “PCLB002” and “PCLB003”. Both codes refer to a single session of BoNT-A injection(s) within striated muscles, with or without electromyographic guidance (003 and 002, respectively).

We further searched the PMSI database to identify patients coded with MS who received at least one BoNT-A injection between 2014 and 2020 for the treatment of NDO. “JDLE900” and “JDLE332” were the used CCAM codes, which both refer to the intradetrusor injection of BoNT-A via cystoscopy. Of note, “JDLE900” was removed from the CCAM codes in 2014.

### 5.3. Outcomes

Among patients with MS, we assessed the prevalence of spasticity, which was defined as the number of patients with MS coded with spasticity between 2014 and 2020 divided by the total number of patients coded with MS during the same period. Other evaluated outcomes in patients coded with MS-related spasticity were the number and proportion of patients who received at least one BoNT-A injection within striated muscles between 2014 and 2020, the number and proportion of patients who received at least three BoNT-A injections between 2014 and 2020, and the time interval between two BoNT-A injections administered within striated muscles. In addition, in order to reduce immortal time bias attributable to retrospectively entered data in the PMSI database, trends in BoNT-A use were evaluated in patients coded with MS-related spasticity who initiated BoNT-A therapy between 2015 and 2020. In order to further reduce bias and confounding, we also estimated the rate of BoNT-A treatment initiation in patients with MS hospitalised for the first time between 2017 and 2020.

Another analysis exclusively focused on patients with MS treated with BoNT-A injections between 2014 and 2020 for NDO. In these patients, we examined the rate of intradetrusor BoNT-A use among the total MS population, the mean number of intradetrusor BoNT-A injections per patient, and the time interval between two intradetrusor BoNT-A injections. We also evaluated the proportion of patients coded with MS who received both BoNT-A injections within the striated muscle and the detrusor smooth muscle between 2014 and 2020, focusing on the overlap of these two types of injections within one-year intervals.

### 5.4. Statistical Analysis

Descriptive statistics were used to report demographic characteristics and study outcomes. Categorical variables were reported as frequency counts with percentages. Mean and SD were calculated for continuous variables. Data were analysed by sex and by age groups of ≤20 years, 20–39 years, 40–59 years, 60–79 years, and ≥80 years. The Chi-square test was used to evaluate trends and relationships between variables. No imputation of missing data was performed. All analyses were performed using the SQL server software (version 2022; Microsoft, Redmond, WA, USA).

### 5.5. Ethics

The study was conducted in compliance with all French laws and regulations as well as the Declaration of Helsinki and Good Clinical Practice guidelines. Data access was approved by the French Data Protection Agency (*Commission Nationale de l’Informatique et des Libertés*, CNIL (Paris, France). This study did not involve direct participation of humans.

## Figures and Tables

**Figure 1 toxins-15-00280-f001:**
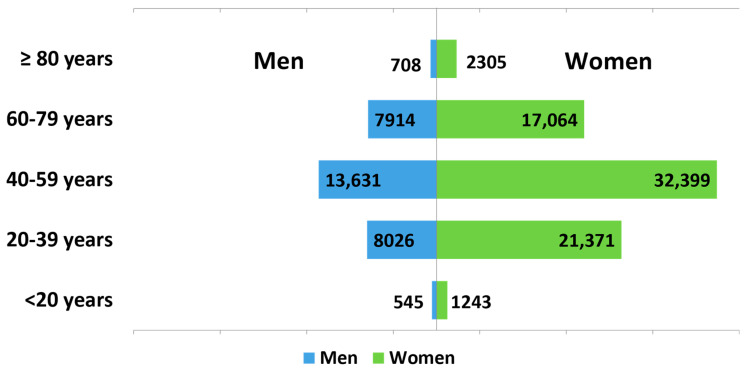
Age and sex distribution among a total of 105,206 patients with multiple sclerosis.

**Figure 2 toxins-15-00280-f002:**
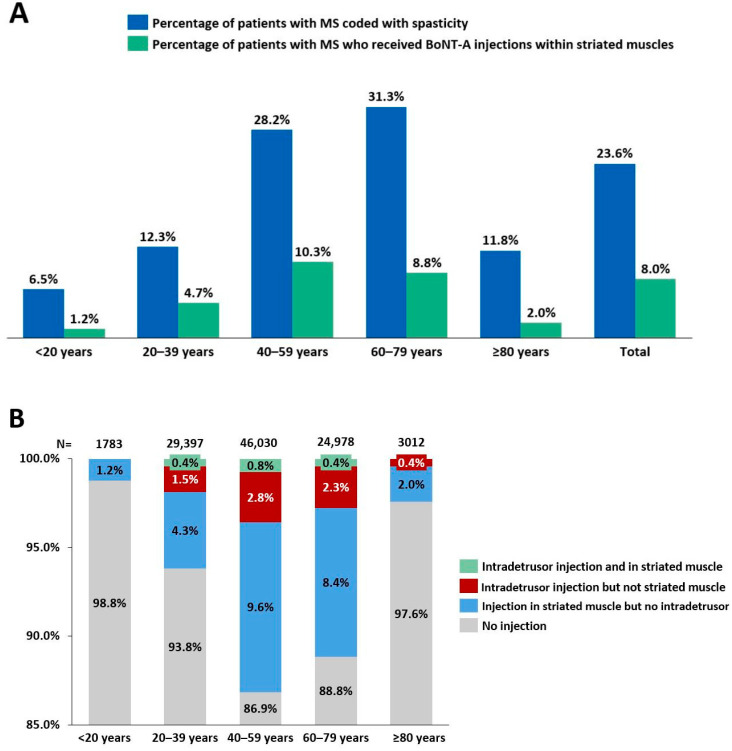
Proportions of patients with multiple sclerosis (MS) coded with spasticity and treated with botulinum toxin type A (BoNT-A) administered within striated muscle (**A**) and within detrusor smooth muscle (**B**) according to age.

**Figure 3 toxins-15-00280-f003:**
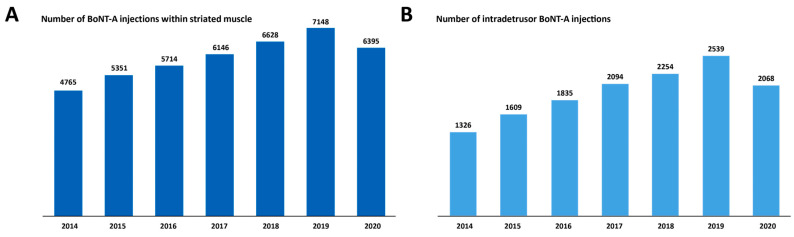
Number of botulinum toxin type A (BoNT-A) injections administered within striated muscle for multiple sclerosis-related spasticity (**A**) and within detrusor smooth muscle for neurogenic detrusor overactivity (**B**) between 2014 and 2020.

**Figure 4 toxins-15-00280-f004:**
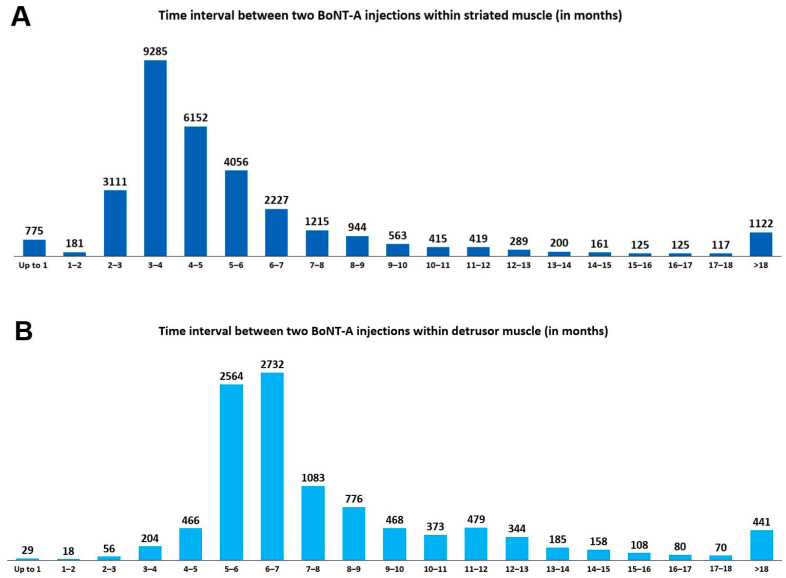
Time intervals (in months) between two botulinum toxin type A (BoNT-A) injections administered within striated muscle (**A**) and within detrusor smooth muscle (**B**).

**Figure 5 toxins-15-00280-f005:**
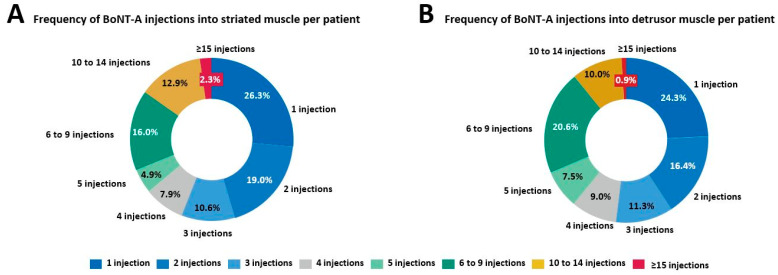
Frequency of botulinum toxin type A (BoNT-A) injections administered within striated muscle between 2015 and 2020 for multiple sclerosis-related spasticity (**A**) and within detrusor smooth muscle between 2014 and 2020 for neurogenic detrusor overactivity (**B**).

**Table 1 toxins-15-00280-t001:** Initiation rate of botulinum toxin type A (BoNT-A) therapy in patients with multiple sclerosis (MS) newly hospitalised between 2017 and 2020.

	First Hospitalisation for MS	Newly Treated Patients with BoNT-A	BoNT-A Treatment Initiation Rate
2017	5935	1064	17.9%
2018	5808	1044	18.0%
2019	5664	1017	18.0%
2020	5019	858	17.1%

## Data Availability

Restrictions apply to the availability of these data since the data underlying this publication were provided by the Agence Technique de l’Information sur l’Hospitalisation (ATIH) under contract to Ipsen.

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
