# Peer review of "Trends in Botulinum Toxin Use among Patients with Multiple Sclerosis: A Population-Based Study"

_toxins, 2023, doi:10.3390/toxins15040280_

Round 1

Reviewer 1 Report

The manuscript, ‘Trends in botulinum toxin use among patients with multiple sclerosis: A population-based study’ highlights the trends of clinical use of BoNT-A treatment trends in patients with MS -within striated muscle and detrusor smooth muscle- between 2014 and 2020 in France. The manuscript is well-composed, and the figures are well-illustrative. In the discussion, the author mentioned about lack of critical information about the dosage and efficacy of repeated treatment. However, this manuscript comprehensive national database of several years to describe BoNT-A use for MS-related spasticity and therefore provides a piece of critical information in the field.

Author Response

Thank you for your positive feedback.

Reviewer 2 Report

I have read with great interest this paper, that describes a population-based retrospective cohort study of trends in botulinum toxin type A treatment in spasticity and neurogenic detrusor overactivity in patients with multiple sclerosis between 2014 and 2020. This is a well written manuscript, and the the current presentation and the data are so interesting.

Author Response

Thank you for your positive feedback.

Reviewer 3 Report

The authors reported an interesting study about the use of Botulinum Toxin among patients with multiple sclerosis in France. I have some comments to the authors:

(i)    In the first part of the introduction, the authors should stress that Botulinum Toxin is routinary used in the treatment of dystonia (without the distinction between cervical dystonia or blepharospasm) and movement disorders in general. Here two important papers that the authors should add in the manuscript and that can help the reader to expand his range of knowledge about BoNT-A:

Anandan C, Jankovic J. Botulinum Toxin in Movement Disorders: An Update. Toxins (Basel). 2021 Jan 8;13(1):42. doi: 10.3390/toxins13010042. PMID: 33430071; PMCID: PMC7827923.

Romano M, et al. Diagnostic and therapeutic recommendations in adult dystonia: a joint document by the Italian Society of Neurology, the Italian Academy for the Study of Parkinson's Disease and Movement Disorders, and the Italian Network on Botulinum Toxin. Neurol Sci. 2022 Dec;43(12):6929-6945. doi: 10.1007/s10072-022-06424-x. Epub 2022 Oct 3. PMID: 36190683.

(ii)   The authors should report the methods right after the introduction.

(iii) I do think that one of the most important limitations of this study is the lack of correlation with other clinical and demographic variables. The authors should clearly state this point in the manuscript and they also should include this among the future directions of the topic.

Author Response

(i) Thank you for your comments. We have now added the two proposed references to the Introduction, along with the following sentence on lines 50 and 51: “Given its potency and favourable safety profile, BoNT-A is routinely used for the treatment of neurologic movement disorders, including dystonia [2,3].”

(ii)  The Instructions for Authors of Toxins (https://www.mdpi.com/journal/toxins/instructions) specify that the Materials and Methods section should be placed at the end of the manuscript right after the Conclusions. 

(iii) As requested, we have now added as a study limitation on lines 227 to 231 that “this study did not examine the correlation between BoNT-A use and clinical and demographic variables, since the use of the anonymised PMSI database does not allow the performance of a cross-check analysis of clinical data. Given the impact of spasticity and NDO on the day-to-day lives of people with MS, studying the clinical and demographic factors that may influence BoNT-A treatment patterns is warranted.”

Reviewer 4 Report

This is a nice study that gives interesting insights into real world data of the administration of botulinum toxin (BoNT) in multiple sclerosis.

A few questions:

1. Do the authors know what was the indication for BoNT injections in spasticity. Is there a standard procedure in France to identify suited patients and to define the treatment goals. Were all injectors familiar with this?

2. What were the subtypes of MS, please provide more details on that (e.g. RRMS, SPMS,PPMS)  It would be interesting to see how often BoNT was used in these subgroups.

3. MS is not a stable disease and spasticity or bladder dysfunction may vary considerably over time. How do the authors deal with this fact?

Author Response

  1. Thank you for your questions. The authors did not know the indications for BoNT injections in spasticity (neither spasticity patterns nor targeted muscles). This information cannot be obtained with the methodology used for data extraction (i.e., PMSI database), given that there are no existing ICD-10 and CCAM codes that identify the indication for BoNT injections in spasticity. Moreover, the PMSI database cannot establish an association between a medical procedure (i.e., BoNT injection) and any disease. There is also no standard procedure in France to identify suited patients and to define the treatment goals. However, we do tend to use the goal attainment scale (GAS) method to define the treatment goals in spasticity management. We believe that if the GAS method was part of the national recommendations for the treatment of spasticity, it could enhance current practice.
  2. In response to your question, we pointed out now on lines 231 to 234 as a study limitation, “we did not evaluate BoNT-A use according to MS subtypes (i.e., relapsing remitting MS, secondary progressive MS, primary progressive MS), as the French version of ICD-10 does not make the distinction between different MS subtypes.”
  3. The adopted methodology in our study (i.e., PMSI database) does not allow any definite answers to be given to this question. However, we agree that the instability of MS and its secondary conditions (spasticity and bladder dysfunction) is an important research topic that requires a dedicated study. Of note, the variation of MS-related spasticity over time was reflected in our age subgroup analysis (Figure 2A), in which the prevalence of MS-related spasticity was found to progressively increase with age, peaking in the 60–79-year age group at 31.3%, before decreasing thereafter. We have also addressed the instability of MS and of its secondary conditions (spasticity and bladder dysfunction) in the Discussion section in the following paragraphs from lines 202 to 221: “[…] there is evidence of decreasing efficacy over time for intradetrusor BoNT-A therapy, particularly after the fourth injection [19,23]. This further reinforces the importance of repeated BoNT-A injections to treat detrusor overactivity and improve bladder compliance in patients with NDO. It has been also suggested that the clinical use of BoNT-A in patients with MS largely depends on symptom burden, rather than merely being a function of the disease duration [13]. Moreover, spasticity may worsen with muscle fatigue, stress, and anxiety [24]. MS-related spasticity also worsens with greater gait disability, as well as with relapses or progression of the disease, treatments for the disease process, and treatments for other symptoms, infections, injuries, or wounds [25]. As such, BoNT-A injections could be scheduled depending on symptom burden as well as the intrinsic and extrinsic factors that can influence spasticity in people with MS. Hence, improving understanding of spasticity triggers and of MS-specific clinical features may enable clinicians to better individualise BoNT-A therapy [24]. The routine assessment of urodynamic parameters (e.g., maximum detrusor pressure, bladder capacity, bladder compliance, non-voiding contractions) is also important for the individualisation of intradetrusor BoNT-A therapy and for improving therapeutic compliance [22,23]”

Reviewer 5 Report

Very interesting and original data based on longitudinal, MS population based study, on french National hospital discharge Database, covering the entire french population, looking at the therapeutic use of BoNT-A.

well analysed and discussed datas.

Author Response

Thank you for your positive feedback.

Round 2

Reviewer 4 Report

Best regards

MN